# Oculomotor freezing reflects tactile temporal expectation and aids tactile perception

Stephanie Badde [1,2✉], Caroline F. Myers [1], Shlomit Yuval-Greenberg [3,4] & Marisa Carrasco[1,2]

The oculomotor system keeps the eyes steady in expectation of visual events. Here, recording microsaccades while people performed a tactile, frequency discrimination task enabled us to test whether the oculomotor system shows an analogous preparatory response for unrelated tactile events. We manipulated the temporal predictability of tactile targets using tactile cues, which preceded the target by either constant (high predictability) or variable (low predictability) time intervals. We find that microsaccades are inhibited prior to tactile targets and more so for constant than variable intervals, revealing a tight crossmodal link between tactile temporal expectation and oculomotor action. These findings portray oculomotor freezing as a marker of crossmodal temporal expectation. Moreover, microsaccades occurring around the tactile target presentation are associated with reduced task performance, suggesting that oculomotor freezing mitigates potential detrimental, concomitant effects of microsaccades and revealing a crossmodal coupling between tactile perception and oculomotor action.

[1] Department of Psychology, New York University, 6 Washington Place, New York, NY 10003, USA. [2] Center for Neural Science, New York University, 6 Washington Place, New York, NY 10003, USA. [3] School of Psychological Sciences, Tel-Aviv University, Ramat Aviv, 6997801 Tel Aviv-Yafo, Israel. [4] Sagol School of Neuroscience, Tel-Aviv University, Ramat Aviv, 6997801 Tel Aviv-Yafo, Israel. ✉email: stephanie.badde@nyu.edu

Temporal predictions enable us to get ready for upcoming sensory events[1,2]. The oculomotor system keeps the eyes steady in expectation of visual stimuli: microsaccades, small fixational eye movements[3–7], are inhibited prior to the onset of temporally predictable visual events[8,9]. We investigated whether this inhibition is restricted to the visual modality or also emerges in touch. The presence of a coupling between temporal expectation in touch and oculomotor freezing would reveal a surprising crossmodal link across perception, cognition, and action. Furthermore, we explored the functionality of anticipatory oculomotor inhibition by assessing whether it aids tactile perception.

Such a crossmodal link opens the possibility that microsaccadic inhibition is a marker of a supramodal mechanism of temporal expectation. Consistently, research in the auditory domain[10] has indicated the presence of coupling between temporal expectation and oculomotor freezing. Yet, sensory information is likely to share a common source across vision and audition; the sound of a colleague's steps in the hallway might reliably predict her visual presence in your office. Hence, the oculomotor system might show a preparatory response for auditory events in expectation of an accompanying visual event. In contrast, given that humans do not assume by default that tactile and visual events share a common cause[11], similar predictions about an upcoming tactile stimulus would not usually trigger visual expectation. Thus, the tactile modality is a strong test case for the possibility of microsaccadic inhibition as a marker of supramodal temporal expectation.

A coupling between oculomotor action and tactile temporal expectation would raise questions about the functionality of anticipatory oculomotor inhibition. Why should the eyes be held steady in expectation of tactile events? Three accounts of anticipatory oculomotor inhibition are plausible: (1) even within the same modality, action and perception can be decoupled[12–16]. Thus, microsaccadic inhibition may be a mere by-product of temporal expectation and does not serve any perceptual purpose. (2) Anticipatory microsaccadic inhibition may specifically aid visual perception by ensuring the absence of microsaccades around the time of the visual event, as they can impair perception of a brief stimulus due to visual blur or masking[6]. In addition, saccadic suppression effects can lead to spatial and temporal distortions in the perception of visual stimuli presented around the onset of saccades[17,18] and microsaccades[19,20]. Consistent with

this visual account, microsaccades during target presentation are associated with impaired performance in a visual temporal expectation task[9]. But according to this account, performance in a nonvisual, tactile task should not be affected by microsaccades before or during tactile target presentation. (3) Anticipatory microsaccadic inhibition may serve perception in general by preventing the withdrawal of processing resources. Single-cell recordings have shown that microsaccades suppress target-related neuronal activity in the superior colliculus[21] and middle temporal as well as ventral and lateral intraparietal areas[22]. Given that all these brain structures receive inputs from multiple senses[23–26], microsaccadic inhibition could help preserve processing resources that aid tactile perception.

To address the potential supramodality and functionality of the link between temporal expectation and oculomotor freezing, we tested whether microsaccadic inhibition (i) reflects tactile temporal expectation and (ii) benefits tactile perception. We manipulated participants' expectation about the onset time of a tactile target vibration using a tactile temporal cue, analogous to the procedure used in visual[9] and auditory[10] tasks. The cue preceded the target stimulus by one of five intervals (1, 1.5, 2, 2.5, or 3 s). This cue–target interval, often called a foreperiod, was either held constant within a block (regular condition), allowing participants to form specific temporal predictions about the precise onset of the tactile target, or varied within a block (irregular condition), allowing only for general temporal predictions about the extended time window during which the target could occur. No information was provided about the tactile cue at all, rendering all learned associations incidental. Participants performed a frequency-discrimination task on the tactile target while fixating straight ahead, their eye movements were continuously recorded (Fig. 1a). For each participant, we compared microsaccade rates across regular and irregular conditions and tested their relation to performance in the tactile-discrimination task.

To preview our results, we find that microsaccades are inhibited prior to tactile targets and more so in regular than irregular blocks, revealing a tight crossmodal link between tactile temporal expectation and oculomotor action. Moreover, microsaccades shortly before, during, and shortly after the onset of the tactile target are associated with slower and incorrect responses, revealing a functional role for oculomotor freezing in tactile perception.

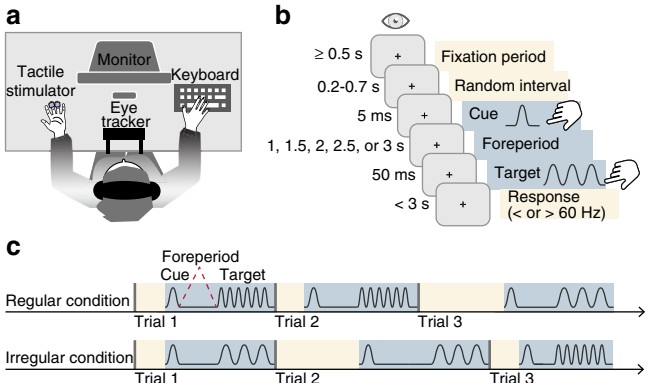

**Fig. 1 Setup, procedure, and design. a** Setup. Participants sat at a table, their head supported by a chin and forehead rest, and fixated straight ahead while their eye position was monitored. Tactile stimulators were attached to the nondominant hand; the dominant hand rested on a keyboard. **b** Trial timeline. Trials began contingent on 0.5 s of continuous fixation, followed by a variable time interval of 0.2–0.7 s, ensuring that the stream of tactile stimuli within any block was nonrhythmic. Tactile cue and target were separated by a foreperiod of 1, 1.5, 2, 2.5, or 3 s. The cue was a single protruding movement of the stimulator tip; the target stimulus was a 50-ms-long vibration. Participants indicated by button press whether they perceived the target frequency as faster or slower than 60 Hz. **c** Design. We manipulated the degree of temporal predictability by either keeping the foreperiod (cue–target interval, blue ribbons) constant—regular condition—or variable—irregular condition—within blocks.

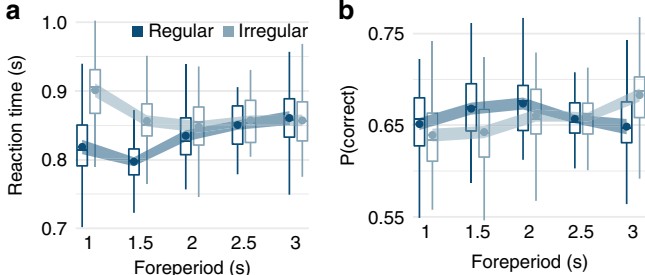

**Fig. 2 Task performance and temporal predictability.** Effects of temporal predictability condition (dark blue, regular; light blue, irregular) and foreperiod (x-axis) on **a** reaction times and **b** proportions of correct responses. Boxplots indicate the distribution of participant-level mean values per condition adjusted by their overall mean (center line, median; box limits, upper and lower quartiles; whiskers, minimum and maximum limited by 1.5× interquartile range). Circular markers show group-level mean values; the width of the ribbon around each marker equals the predictability-condition-adjusted standard error, which indicates the degree of intersubject variation in the difference between regular and irregular conditions, and therefore whether there is an effect of predictability condition on the dependent variable. All statistics are based on the full dataset (N = 30 participants, 100 repetitions per each of the 10 conditions and participant), and source data are provided as a Source Data file.

## Results

**Task performance and temporal predictability**. The behavioral results indicate that our design (Fig. 1c) allowed participants to develop temporal expectations about the onset of the tactile stimulus: the effect of temporal predictability on participants' task performance—response time and accuracy—varied with the length of the cue-to-target foreperiod (Fig. 2; see Supplementary Data 1 for full statistical models, including pairwise and polynomial contrasts). Note that in addition to reaction time, response accuracy was affected even though the difficulty of the task was adjusted throughout the session. Apparently, our adaptive procedure did not sufficiently account for fluctuations in task difficulty due to internal variables such as fatigue; indeed, accuracy remained below the intended level of 71%. Reaction times in the regular condition initially decreased followed by a nonlinear increase; in the irregular condition, they initially decreased and then reached an asymptotic level (Fig. 2). Response accuracy in the regular condition followed an inverted U shape; in the irregular condition, it increased linearly. In trials with short foreperiods, participants' performance—response time and accuracy—was better in regular than irregular foreperiods. For the longest foreperiod, accuracy was higher in irregular than regular foreperiods. In sum, participants developed temporal expectations about the target onset as manifested in both response speed and accuracy.

**Microsaccade frequency and temporal predictability**. Pretarget microsaccadic inhibition. Microsaccade rates reflected tactile temporal expectations: they were reduced prior to the onset of the target vibration, and pretarget microsaccade rates were consistently lower in regular than irregular conditions (Fig. 3a, c, dark vs. light blue, light-gray-shaded area). In all conditions, microsaccade rates were lower in the 200-ms interval just before the onset of the target stimulus than in a comparison interval, 300–500 ms after the cue (Fig. 3a, c, light vs. medium gray-shaded areas, pretarget vs. post cue, see Supplementary Data 2 for full statistical models). The extent of this pretarget microsaccadic inhibition varied with predictability condition and foreperiod. With regular foreperiods, in the earlier post-cue interval, microsaccade rates were higher for shorter than longer foreperiods (Fig. 3b, left panel, medium gray-shaded area, Fig. 3a, c, dark-blue markers and medium gray-shaded area), and they were reduced to a relatively constant low level in the 200-ms interval before target onset (Fig. 3b, left panel, line ends; Fig. 3a, c, dark-blue markers and light-gray-shaded area). In contrast, for irregular foreperiods, microsaccade rates were relatively constant across foreperiods in the post-cue interval (Fig. 3b, right panel, medium gray-shaded area; Fig. 3a, c, light-blue markers and medium gray-shaded area), but they declined with

increasing foreperiods in the pretarget interval (Fig. 3a, c, light-blue markers and light-gray-shaded area). In sum, microsaccades were inhibited prior to an expected tactile target stimulus, and the degree of inhibition systematically varied with the degree of temporal predictability.

Post-target microsaccadic inhibition. Microsaccade rates varied with the degree of tactile temporal expectations even after the tactile target had been presented. They were reduced 0–200 ms after the offset of the target vibration compared with the pretarget interval (Fig. 3a, c, light vs. dark-gray-shaded areas, pretarget vs. post target), and post-target microsaccade rates were consistently lower in regular than irregular conditions (Fig. 3a, c, light vs. dark-blue markers, dark-gray-shaded area).

**Microsaccades and task performance**. To assess the possible functionality of oculomotor freezing for tactile perception, we investigated the relation between microsaccade rates and tactile task performance. Microsaccade rates in a temporal cluster ranging from approximately 200 ms prior to target onset to 200 ms after the target offset were lower in trials with fast than slow responses (Fig. 4a), and in trials with correct than those with incorrect responses (Fig. 4b). Moreover, responses were faster and more accurate with increasing time intervals between the last microsaccade and the tactile target (Fig. 4c, reaction time: $\chi^2(1) = 11.51$, $p < 0.001$, $\beta = 0.009$, CI = (0.001 to 0.018); response accuracy: $\chi^2(1) = 4.86$, $p = 0.028$, $\beta = -0.061$, CI = (−0.111 to −0.011)), independent of predictability condition (interaction, reaction time: $\chi^2(1) = 0.23$, $p = 0.634$, $\beta = 0.002$, CI = (−0.009 to 0.013); response accuracy: $\chi^2(1) = 1.43$, $p = 0.232$, $\beta = 0.039$, CI = (−0.032 to 0.109)). In addition, in these trials with a microsaccade within 1 s before the target stimulus, there was an additive main effect of predictability condition in reaction times (reaction time: $\chi^2(1) = 62.30$, $p < 0.002$, $\beta = 0.047$, CI = (0.035 to 0.058); response accuracy: $\chi^2(1) = 0.00$, $p = 0.319$, $\beta = -0.036$, CI = (−0.107 to 0.034)), indicating faster responses for regular than irregular conditions. Furthermore, single-trial reaction times and response accuracy were impaired in the presence of a microsaccade during the target vibration (Fig. 4d, center panel; reaction times: $\chi^2(1) = 5.12$, $p = 0.024$, $\beta = -0.011$, CI = (−0.020 to −0.001); response accuracy: $\chi^2(1) = 4.49$, $p = 0.034$, $\beta = 0.057$, CI = (0.004 to 0.109)), as well as in the 200-ms interval before (Fig. 4d, left panel; reaction times: $\chi^2(1) = 15.36$, $p < 0.001$, $\beta = 0.020$, CI = (0.010 to 0.030); response accuracy: $\chi^2(1) = 4.67$, $p = 0.031$, $\beta = -0.063$, CI = (−0.120 to −0.006)) and after (Fig. 4d, right panel; reaction times: $\chi^2(1) = 4.13$, $p = 0.042$, $\beta = -0.014$, CI = (−0.027 to 0.001); response accuracy: $\chi^2(1) = 2.46$, $p = 0.117$, $\beta = -0.052$, CI = (−0.118 to 0.013)) the target. We note

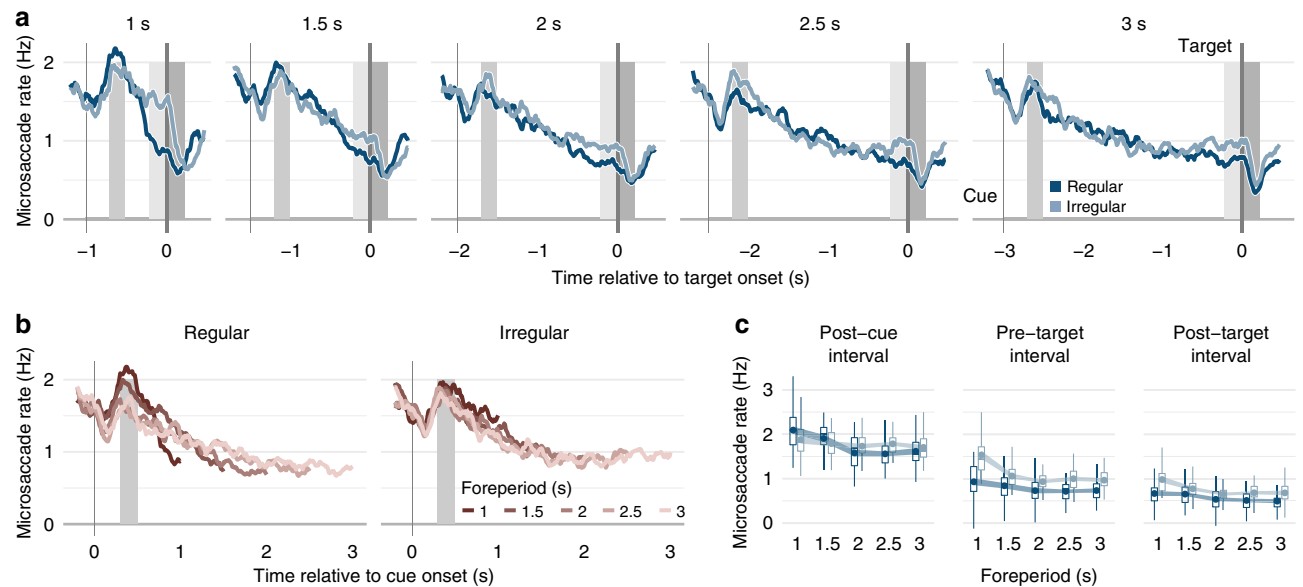

**Fig. 3 Microsaccade frequency and temporal predictability. a** Group-average microsaccade rates as a function of trial time relative to the onset of the tactile target stimulus separately for each predictability condition (dark blue, regular; light blue, irregular) and foreperiod (panels). Shaded vertical bars indicate the cue and target stimulus (blackish gray), shaded rectangles the post-cue (medium gray), pretarget (light gray), and post-target (dark gray) intervals. **b** Group-average microsaccade rate timelines relative to the cue onset separately for each predictability condition (panels) and foreperiod (red shades). **c** Microsaccade rates in a comparison interval 300–500 ms after the onset of the tactile cue (left panel), in the 200-ms interval before the onset of the tactile target stimulus (center panel), and in a post-target interval 0–200 ms after the offset of the tactile stimulus (right panel), separately for each predictability condition and foreperiod (x-axis). Boxplots indicate the distribution of participant-level mean values per condition adjusted by their overall interval mean (center line, median; box limits, upper and lower quartiles; whiskers, minimum and maximum limited by 1.5× interquartile range). Circular markers show group-level mean values; the width of the ribbon matches the predictability-condition-adjusted standard error that indicates the degree of intersubject variation in the difference between regular and irregular conditions, and therefore whether there is a significant effect of predictability condition on microsaccade rates. All statistics are based on the full dataset (N = 30 participants, 100 repetitions per condition and participant), and source data are provided as a Source Data file.

that this negative impact of microsaccades on performance was not a function of stimulus frequency, as the effect on behavioral performance was present shortly before, during, and shortly after the tactile target. No significant correlation emerged between participants' average microsaccade rate across the trial and task performance (Supplementary Fig. 1; reaction time: $r = 0.20$, $p = 0.295$, CI = (−0.35 to 0.37); response accuracy: $r = 0.01$, $p = 0.940$, CI = (−0.17 to 0.52)), and neither reaction times nor response accuracies depended significantly on microsaccade directions (Supplementary Fig. 2; reaction times: $\chi^2(1) = 2.66$, $p = 0.103$, $\beta = 0.002$, CI = (−0.004 to 0.004); response accuracy: $\chi^2(1) = 0.35$, $p = 0.554$, $\beta = −0.001$, CI = (0.009 to 0.16)).

## Discussion

In the current study, microsaccades were recorded while people performed an unrelated tactile task, which enabled us to test whether the oculomotor system shows a preparatory response for tactile events by keeping the eyes steady. We varied the temporal predictability of the tactile targets by presenting them either at a constant, and therefore highly predictable time point following a tactile cue, or at a pseudovariable time point after the cue, which allowed only for general temporal predictions. Several findings emerged: microsaccades are always inhibited prior to the onset of the tactile target—microsaccade rates decrease from the "post-cue" to the "pre-target" interval—and more so preceding precisely predictable targets. Hence, remarkably, our study reveals that oculomotor freezing reflects temporal expectation in the tactile modality. Given that humans do not assume by default that tactile and visual events share a common cause[11], the tactile modality provides a strong test case that microsaccadic inhibition is a

marker of supramodal temporal expectation. Moreover, microsaccades that occurred shortly before, during, or shortly after the target vibration are associated with reduced task performance in tactile discrimination, supporting a functional role of anticipatory microsaccadic inhibition and revealing a crossmodal coupling between the oculomotor system and tactile perception. Anticipatory microsaccadic inhibition could help preserve processing resources that aid tactile perception. This finding in the tactile modality indicates that the functional benefit of microsaccadic inhibition goes beyond vision[8,9] and extends across modalities.

Anticipatory oculomotor freezing: a marker of supramodal temporal expectation. This study reveals that expected tactile stimuli are preceded by microsaccadic inhibition, and that the degree of inhibition increases with temporal predictability. The presence of a relation between temporal predictability and oculomotor freezing in touch provides compelling evidence that microsaccadic inhibition reflects temporal expectation independent of modality, as tactile events are unlikely to trigger visual expectations. These findings, in combination with identical and similar effects in audition[10] and vision[8,9,27], suggest that microsaccadic inhibition can be a marker of a supramodal mechanism of temporal expectation. Consistently, oscillations reflecting visual–tactile temporal expectation have sources over motor rather than sensory cortices[28]. The possibility that temporal expectations are supramodal is in line with the definition of expectation as regarding the prior probability for an event independent of the event's task relevance. The distinction between expectation and attention (related to the event's task relevance, which was constant in both conditions in this study) in the temporal domain[27,29,30] is based on the well-established distinction between expectation and attention in both the spatial[2,31,32] and the feature[2,33]

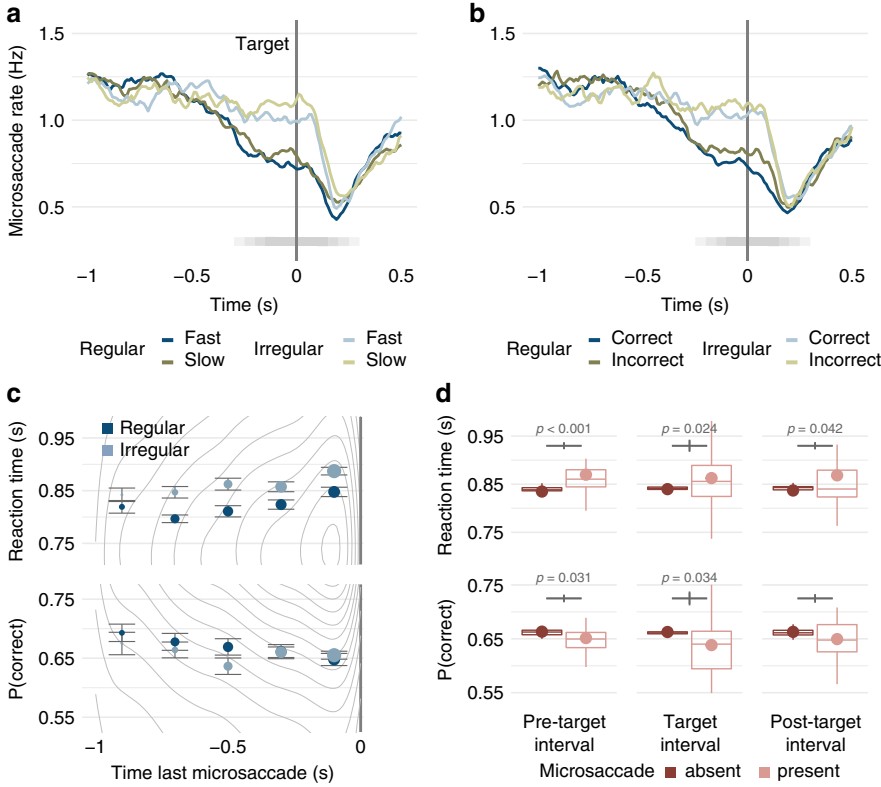

**Fig. 4 Microsaccades and task performance. a**, **b** Microsaccades by task performance. Group-average microsaccade rates as a function of time relative to target onset (vertical gray line) split by **a** reaction times (blue: fast, green: slow responses) and **b** response accuracy (blue: correct, green: incorrect responses) in regular (dark shade) and irregular (light shade) blocks. Temporal clusters with significant differences in microsaccade rates (RT: $p < 0.001$, accuracy: $p < 0.001$, two-sided cluster permutation tests) between performance categories are indicated by shaded horizontal bars. Each tile corresponds to a 200-ms bin with a significant difference ($p < 0.05$, two-sided permutation tests); darker shades indicate overlap between bins. **c** Task performance by microsaccade latencies. Gray lines indicate the 2d frequency distribution of single-trial reaction times (upper panel) and correct responses (lower panel) as a function of the latency of the last microsaccade before the target stimulus. Circular markers show group mean reaction times in regular (dark blue) and irregular (light blue) conditions for 200-ms-long time bins. Marker size represents the percentage of trials per bin. Error bars indicate standard errors corrected for between-participant variability. **d** Task performance by microsaccades. Reaction times (upper panel) and proportion correct (lower panel) split by the presence of microsaccades in the pretarget, target, and post-target interval (dark red, absent; light red, present). Boxplots indicate the distribution of participant-level mean values per condition adjusted for their overall mean (center line, median; box limits, upper and lower quartiles; whiskers, minimum and maximum limited by 1.5× interquartile range). Note that trials without a microsaccade in the respective interval were more frequent than those with a microsaccade resulting in a narrower distribution. Circular markers show group-level mean values. Vertical gray lines indicate the standard error of the difference between conditions; $p$ values are based on generalized linear mixed models predicting single-trial performance from the presence of a microsaccade (see "Results" for full statistical information). All statistics are based on the full dataset ($N = 30$ participants, 100 repetitions per condition and participant) and weighted by the number of trials per condition. Source data are provided as a Source Data file.

domains. Supporting the distinction in the temporal domain, modulations of evoked potentials caused by visual–tactile temporal expectation precede those caused by modality-specific attention[34]. Furthermore, it seems that whereas temporal expectation is likely supramodal, temporal attention is maybe not; crossmodal transfer of temporal attention[35] does not emerge when controlling for unspecific temporal expectation[36]. The strong relation between microsaccade rates and temporal expectation is in line with the idea that microsaccades reflect perceptual and cognitive states[3–6], in addition to their role in visual perception[6]. Microsaccade direction can indicate visual[37–39] and auditory[40] spatial attention, and distinguish between distributed and local visual processing[41]. Microsaccade rates decrease with temporal attention to visual targets[27], increased attentional[38] and working memory[42] load, as well as task difficulty in visual[43] and nonvisual[44,45] tasks.

Poststimulus oculomotor freezing and tactile temporal expectation. This study also shows that tactile stimuli, both cue and target, induce subsequent oculomotor freezing. Microsaccade rates decreased in response to tactile cues and target stimuli—a

phenomenon known in vision and audition as poststimulus microsaccadic inhibition[37,46]. The degree of microsaccadic inhibition following the tactile target stimulus increased with temporal predictability. There are two, non-mutually exclusive, plausible reasons for this modulation: (1) a simple consequence of microsaccadic inhibition prior to the target presentation, which was also stronger with higher temporal predictability and (2) a marker, per se, reflecting persistent effects of temporal expectation during perceptual processing. Consistent with the latter, various perceptual and cognitive processes influence the degree and time course of poststimulus microsaccadic inhibition (in the absence of prestimulus microsaccadic inhibition). For instance, poststimulus microsaccadic inhibition is a reliable indicator of conscious visual perception[47,48], visual and auditory oddball detection[49,50], and auditory categorization[51]. Together, these studies and our results in the tactile modality indicate that, in addition to pretarget microsaccadic inhibition, poststimulus microsaccadic inhibition can also be a marker of supramodal temporal expectation.

**A functional role for oculomotor freezing in tactile perception.** The present study reveals that microsaccades preceding, accompanying, and following the tactile target stimulus are associated with impairments of task performance. This crossmodal link between oculomotor behavior and tactile perception unveils a functional role for microsaccadic inhibition in tactile discrimination. How can microsaccades occurring before, during, or even shortly after the target play a role in tactile perception? We do not know, but there are two plausible, interrelated answers, namely, microsaccades interact with (1) neuronal and (2) cognitive resources. (1) Microsaccades suppress target-related neuronal activity in the superior colliculus[21], middle temporal, and intraparietal areas[22] for a few hundred milliseconds. In turn, intraparietal areas are highly active during tactile vibration discrimination[52], suggesting that a microsaccade-driven withdrawal of neural resources in these areas[22] could affect stimulus processing and thus contribute to the observed decrement in performance in the tactile task. (2) As discussed above, microsaccadic inhibition increases with increasing task demands[38,42–44], raising the possibility that microsaccadic inhibition frees cognitive resources. Given that tactile vibration discrimination profits from cognitive resources such as spatial attention[53], this possibility could also account for the observed negative interaction between microsaccades and tactile task performance. These two likely answers are not mutually exclusive, as those brain areas that exhibit microsaccadic suppression of task-related neural activity in the superior colliculus[21] and middle temporal and intraparietal areas[21,22] are also associated with the allocation of attention[54,55]. Further, both possibilities are consistent with the finding that microsaccades interrupt the evidence accumulation process in perceptual decision-making[56]. The effect of microsaccades on neural or cognitive resources could explain the observed relation between microsaccadic inhibition and task performance while providing a possible explanation for the relation between oculomotor freezing and temporal expectation: mitigation of negative effects concomitant to microsaccades. We note that in addition to these two possibilities, a common factor (e.g., alertness) could unidirectionally influence both microsaccades and tactile perception rather than interact or even covary with microsaccade occurrences (which would be consistent with (2)). Thus, it is possible that microsaccadic inhibition is not intrinsically functional but a by-product of temporal expectation and thus accidentally associated with improved task performance. Yet, the effect of microsaccades on tactile task performance appears to be constant across temporal predictability conditions, suggesting that the relation between microsaccades and tactile perception might be independent of temporal expectation. We also note that all statistical models used here operate on single-trial basis, and no correlation emerged between participants' overall microsaccade rate and their task performance. Thus, at least some extraneous variables with antagonistic effects on microsaccadic inhibition and tactile task performance that vary across individuals rather than individual trials (e.g., fatigue[57] and familiarity[58]) can be excluded as a source of the effect. In this tactile study, performance was impaired when microsaccades occurred shortly before, during, or shortly after the tactile target. Given that a performance benefit of pretarget microsaccadic inhibition has emerged for visual[38,59,60] and visual–auditory[61] tasks, we conclude that the link between microsaccades and perception could be a supramodal functional phenomenon. Given that brain structures associated with microsaccades, such as superior colliculus[21] and middle temporal as well as ventral and lateral intraparietal areas[22], receive inputs from multiple senses[23–26], microsaccadic inhibition might help preserve processing resources that aid perception in different modalities.

**Time specificity of temporal expectation effects.** The effects of temporal predictability on tactile task performance indicate difficulties to profit from precise temporal information at longer time intervals and a limited use of hazard rates. Behavioral benefits that resulted from the possibility to make specific temporal predictions in regular blocks were restricted to short foreperiods. Non-mutually exclusive explanations have been advanced: (1) the range of time periods during which humans can profit from specific temporal predictability might be limited because timing uncertainty increases with longer intervals[9,62,63]. Consistently, in regular blocks with the possibility for precise timing, the response time effect peaked around 1–1.5 s and the accuracy peaked at 1.5–2 s. (2) Given that there is only general temporal predictability with irregular foreperiods, the probability for the target onset increases with time[9,64]. The finding that humans often utilize hazard rates to prepare for upcoming events[1] predicts a nonlinear increase of performance with foreperiod duration in irregular blocks. This pattern emerges for response accuracies, but the benefit in reaction times reached an asymptotic level far above the minimum reaction times in regular blocks, suggesting that participants allocated the additional preparatory advantage gained from hazard rates to response accuracy. In contrast to task performance, pretarget microsaccade rates in regular blocks reflect the availability and use of precise temporal information by the oculomotor system, even at long foreperiods. In further contrast to task performance, pretarget microsaccade rates in irregular blocks were not in agreement with the use of hazard rates. In regular blocks, microsaccade rate timelines declined at different speeds for different foreperiods, and before target onset, they reached a low rate practically independent of foreperiod (Fig. 3b). The impressive match of the steepness of the decline to the duration of the foreperiod indicates the availability of precise temporal information across all durations. In contrast, the speed with which microsaccade rate timelines decline in irregular blocks is similar across foreperiods, and microsaccade rate timelines seem to asymptote in their decline at around 2 s after the cue (Fig. 3b). This asymptotic pattern in irregular blocks is inconsistent with the use of hazard rates for microsaccadic inhibition, as the asymptotic microsaccade rates were higher than those in regular blocks. We suggest that microsaccadic inhibition is costly[27,65], and that this cost increases with longer periods of inhibition. If the cost increases faster over time than the possible advantage from the hazard rate, tailoring microsaccadic inhibition to the average foreperiod of 2 s could be resource-optimal.

**Scheduled microsaccade generation.** Remarkably, the evolution of microsaccade rates over time revealed in this study raises the possibility that microsaccades are generated in advance, as preparatory compensation for subsequent microsaccadic inhibition. The intensity of the rebound after post-cue microsaccadic inhibition (Fig. 3b, left panel, gray shaded area) varied as a function of foreperiod in regular blocks, when participants were able to precisely predict the target onset. Microsaccades during the post-cue rebound were most frequent before the onset of rapidly increasing inhibition in trials with short foreperiods. It seems possible that microsaccades are preemptively triggered to balance the number of microsaccades in the next few seconds. Importantly, the restriction of this phenomenon to regular blocks could not be explained by the difference in pacing between the different foreperiods (Supplementary Fig. 3). This limitation to trials with high temporal predictability suggests that precise temporal planning is needed to proactively schedule microsaccades. Usually, microsaccade triggering is characterized either as automatic, elicited by neural noise[66–68], or as reactive, e.g., driven by current changes in visual input[69–71], top–down signals from spatial attention[39,72,73] (but see ref. [74]), or accumulated fixation error signals[46,72]. A proactive component has not been included yet in models of microsaccade generation. Future models should take into account the observation that microsaccades are possibly generated in anticipation of an upcoming period of microsaccadic inhibition.

In conclusion, this study reveals a tight crossmodal coupling between oculomotor action and tactile temporal expectation,

which portrays anticipatory oculomotor freezing as a marker of supramodal temporal expectation. Moreover, microsaccades are shown to be associated with reduced task performance, indicating a functional role for microsaccadic inhibition, and revealing a surprising crossmodal link between miniature eye movements and tactile perception.

## Methods

**Participants.** Thirty persons recruited at New York University (26 right-handed, 10 male, 19–37 years old, mean 27 years) participated in the experiment. Sample sizes were increased by 50% compared with other studies using the same experimental protocol[9,10], as our analyses related single-trial response times and accuracy to microsaccade occurrences. No participant was excluded. One additional participant could not complete the study due to problems to maintain fixation. All participants reported normal or corrected-to-normal vision, and absence of tactile as well as motor impairments. Twenty-eight participants were naive with respect to the purpose of the study. They received course credit or a small monetary compensation. The study was approved by the internal review board of New York University's Psychology Department, and the experiment was conducted in accordance with the general guidelines in the Declaration of Helsinki. Participants gave written informed consent prior to the beginning of the experiment.

**Apparatus and stimuli.** Participants sat at a table, resting their hands on the table surface. Their head was supported by a chin and forehead rest (Fig. 1a). Tactile stimulators (plectrum piezo stimulators, Dancer Design, St. Helens, UK) were attached to the dorsal side of the distal ring and middle fingers of the nondominant hand. Tactile stimulation consisted of a cue stimulus, a single, 10-ms-long, protruding movement of the tip of the stimulator, and a target stimulus, a 50-ms-long vibration ranging from 30 to 90 Hz, created by a sinusoidal, protruding movement of the tip of the stimulator. Cue and target were always applied to the same finger. Before each session, the stimulation intensity of the stimulators was adjusted so that the perceived stimulus strength was distinctly suprathreshold and subjectively matched across fingers.

A black fixation cross was centrally displayed on gray background at 57-cm distance from the participant (Fig. 1a). Eye position was monitored online and recorded at 1000 Hz using an infrared eye-tracking system (Eyelink, SR Research, Ottawa, Canada). Participants wore headphones playing white noise to shield off any auditory cues produced by the tactile stimulators. The experimental program was written in Matlab (The Mathworks, Natick, MA, USA) and used the Psychophysics Toolbox extension[75], which interfaced with the tactile stimulators via a digital analog converter (National Instruments, Austin, TX, USA).

**Design.** Trials varied with respect to the length of the foreperiod, the time period between cue and target vibration (1, 1.5, 2, 2.5, and 3 s). The foreperiod was either constant (regular condition) or variable (irregular condition) across trials within a block. These two conditions were varied between blocks (Fig. 1c).

**Task.** Participants indicated whether they perceived the tactile target stimulus as faster or slower than their internal standard of a 60-Hz tactile vibration. They used one of two fingers of the nonstimulated hand to press the corresponding button ("<" slower, ">" faster).

**Procedure.** At the beginning of each session, participants completed 20 practice trials discriminating target vibrations of either 80 or 40 Hz. If participants successfully learned to categorize the target vibrations as faster or slower than 60 Hz, i.e., if the last 5–10 responses were correct, task difficulty was adaptively increased; otherwise, the practice trials were repeated. During this initial stimulus adjustment period, the absolute difference between the target stimulus frequency and 60 Hz was chosen based on the 1-up/2-down rule that converges to 71% correct responses[76]; the sign of the frequency difference was chosen randomly. This stimulus adjustment period ended after 10 reversals of the direction of adjustment, ensuring that participants were extensively familiarized with the task before the actual experiment began. To keep task difficulty constant, vibration frequencies were set based on the 1-up/2-down rule throughout the experimental session, across foreperiods and conditions. Feedback was provided only during practice trials.

The beginning of a trial was contingent on participants maintaining fixation for 500 ms (Fig. 1b). The tactile cue was presented after a random interval of 200–700 ms, ensuring that the stream of tactile stimuli was nonrhythmic across trials (Fig. 1c, yellow ribbons). The tactile target stimulus followed the cue stimulus after a variable foreperiod (see "Design"). Reaction times were limited to 3 s, and the next trial started immediately after the response had been registered.

Participants were informed about the mandatory fixation period at the beginning of a trial and encouraged to respond as accurately and fast as possible; no other information was provided to them.

Participants completed 10 blocks of 100 trials each. Each of the five foreperiods was presented 100 times, either all repetitions of one foreperiod in the same block (regular condition) with foreperiods randomly shuffled across blocks, or all foreperiods presented at equal rates and in pseudorandom order within blocks (irregular condition). Each frequency category of the target vibration (faster or slower than 60 Hz) was presented equally often, in pseudorandom order across trials. Blocks with regular and irregular foreperiods were presented alternately; condition order was counterbalanced across participants. To avoid tactile adaptation, we alternated the stimulated finger across blocks while counterbalancing the finger stimulated in the first block across sessions and participants so that different fingers were associated with regular and irregular blocks across sessions. Participants completed the experiment in 2–3 sessions of self-determined length, conducted on different days; overall the experiment took about 120 min.

**Microsaccade detection.** Eye positions were transformed into degrees of visual angle (dva) using a five-point grid alignment procedure. Saccades were detected using a velocity-based algorithm[69] applied to the high-pass filtered time series of eye positions. Saccades were defined as at least 6 consecutive time points with a two-dimensional velocity of at least 6SD above the average velocity per trial; microsaccades were defined as saccades with an amplitude smaller than 1 dva[6]. Saccades and microsaccades identified by the algorithm fell along the main sequence, i.e., saccade amplitudes and peak velocities were highly correlated, Pearson's $r = 0.90$, $p < 0.001$, CI: (0.899–0.901) (Supplementary Fig. 4A). Microsaccade directions were not significantly biased toward either side, that of the stimulated or the response hand, $t(29) = 0.12$, $p = 0.906$, $d = 0.02$ (Supplementary Fig. 4B).

**Analysis of task performance and temporal predictability.** Trials with missing responses (1.8% of all responses), responses shorter than 100 ms or more than 2.5 standard deviations longer than the participant's mean response time (1.1% of all responses) were excluded from all analyses. Reaction times for correct responses were analyzed as the primary dependent variable. To ensure that no speed-accuracy trade-offs occurred, we also analyzed accuracy, which by design should be similar across conditions, as it was adjusted throughout the experimental sessions.

We assessed the effects of temporal predictability on reaction times by fitting a generalized linear mixed model with a gamma distribution and log-link function. The model predicted single-trial reaction times from the temporal predictability condition (regular and irregular) and the length of the foreperiod (1, 1.5, 2, 2.5, and 3 s) while estimating random intercepts for each participant. In addition, we predicted single-trial response accuracies from both factors using a generalized linear mixed model with binomial distribution family and log-link function, i.e., a hierarchical logistic regression. Significant interactions were followed up by contrast analyses (i) comparing performance across the two predictability conditions separately for each foreperiod and (ii) testing for polynomial trends of foreperiod separately for each predictability condition.

**Analysis of microsaccades and temporal predictability.** Trials with blinks (9.6% of all trials) or saccades larger than 1 dva (4.2% of all trials) within a time interval ranging from 1000 ms before to 200 ms after the target stimulus were excluded from all eye data analyses. The same pattern of results emerged when saccades of all amplitudes were included (Supplementary Fig. 5).

To evaluate microsaccadic inhibition preceding the target, microsaccade frequencies in the time window 200 ms prior to target onset were evaluated and compared with microsaccade frequencies in a 200-ms-long time window starting 300 ms after the cue. Note that the same result pattern emerges for slightly longer or shorter intervals (i.e., ±100 ms, Supplementary Fig. 6). To do so, we fitted a hierarchical Poisson model[77] with predictors: predictability condition (regular and irregular), foreperiod (1, 1.5, 2, 2.5, and 3 s), and interval (pre-target and post-cue), and single-participant intercepts to single-trial microsaccade counts in each interval. Significant interactions were again resolved using contrast analyses (i) comparing microsaccade counts across predictability conditions separately for each foreperiod and time interval, (ii) across time intervals separately for each predictability condition and foreperiod, and (iii) testing polynomial trends of foreperiod separately for each predictability condition and time interval.

To evaluate microsaccadic inhibition subsequent to the target presentation, microsaccade frequencies in the 200-ms-long time window after the target offset were evaluated and compared with microsaccade frequencies in the pretarget interval. To do so, we fitted a hierarchical Poisson model with predictors: predictability condition (regular and irregular), foreperiod (1, 1.5, 2, 2.5, and 3 s), and interval (pre-target and post-target), and participant-specific intercepts to single-trial microsaccade counts and followed by the same contrast analyses as for the pretarget interval.

**Analysis of microsaccades and task performance.** We assessed the relation between microsaccadic inhibition and task performance from both possible perspectives[22].

First, we tested whether microsaccade frequencies varied with behavioral performance. To do so, we separately derived microsaccade rate timelines for trials with fast and slow responses (determined by a within-participant median split) and trials with correct and incorrect responses. We assessed differences in microsaccade

frequencies between the timelines using hierarchical Poisson models. These were applied to microsaccade counts within a 200-ms-long sliding window that progressed in steps of 50 ms. Adjacent time windows with significant differences in microsaccade counts between trial types formed a temporal cluster. We tested the largest temporal cluster against a null distribution that was derived using permutations of the condition labels within participants (cluster permutation tests[12,27,78]). As no significant interaction with predictability condition emerged, this analysis was performed exclusively on the main effects of task performance.

Second, we tested whether behavioral performance varied with microsaccade occurrences. For each trial, we extracted the time point of the last microsaccade before target onset and predicted both performance measures from these microsaccade latencies (implemented as scaled covariate) and predictability condition. We fitted single-trial reaction times using a generalized linear mixed model with gamma-distribution family and log link, and single-trial response accuracies using a generalized linear mixed model with binomial distribution family. Furthermore, to assess the effects of microsaccades during the presentation and perception of the tactile target, we compared single-trial accuracies and reaction times in trials that overlapped with the 50-ms target presentation and trials without a microsaccade in this interval. To do so, we conducted generalized linear mixed models with gamma-distribution family and log link for single-trial reaction times, and binomial distribution and logit link for single-trial accuracies. In addition, we assessed whether participants' task performance varied with the number of microsaccades during the pre- and post-target intervals. We predicted single-trial reaction times and response accuracies separately from microsaccade counts within an interval of 200 ms before stimulus onset and an interval 200 ms after the stimulus offset. To do so, we conducted generalized linear mixed models with gamma-distribution family and log link for single-trial reaction times, and binomial distribution and logit link for single-trial accuracies.

To test for a relation between participants' overall microsaccadic frequency and their average task performance, we calculated the Pearson correlation coefficient for each participant average microsaccade rates during the 1-s interval preceding the tactile target and their average reaction time as well as the proportion of correct responses.

Finally, we tested for a relation between the direction of microsaccades during the foreperiod and task performance. We recoded microsaccade direction according to the location of the stimulated and response hand, grouped directions in 90° bins, and used it as a predictor in a generalized linear mixed model with gamma-distribution family and log link for single-trial reaction times, and binomial distribution and logit link for single-trial accuracies.

Significance was affirmed at $p < 0.05$, two-sided, and $p$ values were adjusted to account for alpha inflation according to Holm's procedure[79], or using cluster permutation tests[78] for the analysis of temporal clusters. Contrast analyses were performed conditional on significant interactions. Summation contrasts were assigned to categorical predictors, polynomial contrasts to ordered categorial predictors, and numerical predictors were included as covariates. We report contrast weight estimates ($\beta$) and their 95% confidence intervals on the respective model's scale.

**Reporting summary**. Further information on research design is available in the Nature Research Reporting Summary linked to this article.

## Data availability
The datasets generated during and analyzed during the current study are available in an open-science framework repository with the identifier https://doi.org/10.17605/OSF.IO/7ZSRQ. The source data underlying Figs. 2–4 as well as Supplementary Figs. 1–6 are additionally provided as Source Data file. A reporting summary for this article is available as a Supplementary Information file. Source data are provided with this paper.

## Code availability
Experimental and analysis code are available online: https://doi.org/10.17605/OSF.IO/7ZSRQ. Source data are provided with this paper.

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

## Acknowledgements

This study was funded by the National Institutes of Health R01-EY019693 and NEI R21-EY026185 to M.C. and by the United States–Israel Binational Science Foundation grant 2015201 to S.Y.G and M.C. We thank Dekel Abeles and members of the Carrasco Lab, especially Rachel Denison, Nina Hanning, and Mariel Roberts for helpful comments on the paper, and Karen Tian for help with figure preparation.

## Author contributions

Conceptualization: S.B., S.Y.G., and M.C.; Methodology: S.B., S.Y.G, and M.C.; Software: S.B.; Formal analysis: S.B.; Investigation: S.B. and C.F.M.; Resources: M.C.; Data curation: S.B. and C.F.M.; Writing—original draft: S.B. and M.C.; Writing—review and editing: S.B., C.F.M., S.Y.G., and M.C.; Visualization: S.B.; Supervision: M.C.; Project administration: S.B. and M.C.; Funding acquisition: M.C. and S.Y.G.

## Competing interests

The authors declare no competing interests.
