## [Peer Review File · Nature Communications]

REVIEWER COMMENTS

Reviewer #1 (Remarks to the Author):

This is an interesting manuscript, probably deserving high-impact publication. They that tactile frequency discriminations are accompanied by a slowing of micro-saccades, especially when the stimulus is predictable. I think this is important for several reasons, not least that micro-saccade “freezing” could become a marker for perceptual expectancy. As far as I can tell, the experiments have been conducted with psychophysical rigor, and the analysis appropriate. I make a couple of comments, but in general recommend publication.

As far as I can understand, this is one manuscript of a pair, both showing cross-modal action of micro-saccade slowing, one for touch the other for audition. While I approve high-impact publication of one, both are not high-impact. Either the authors should combine both into a single paper (condensing much of the redundancies), or they publish one as the high-impact, and the other as a follow-up, confirmation with incremental added value, showing the same phenomenon occurs also in audition.

In any event, the manuscript would profit from being condensed, removing repetition and tightening the language. I think a shorter manuscript would be more impactful.

Other than that, nice work!

Reviewer #2 (Remarks to the Author):

The authors investigated the phenomenon of “oculomotor freezing” (inhibition of micro-saccades) as a correlate of temporal expectations. Critically, in contrast to prior work on this phenomenon (most of which by the authors), such freezing was here demonstrated not in anticipation of upcoming vision, but upcoming touch (somatosensory frequency perception). This suggests that this phenomenon may not serve the purpose of visual stabilisation (as previously interpreted), but rather reflect some global correlate of a supramodal temporal expectation. In further support of this, the authors were also able to relate microsaccade rate and timing to somatosensory perception performance; with worse performance when microsaccades occurred close in time to the tactile task, and in trials in which microsaccades occurred at a higher rate.

This is a nice manuscript that is well written and structured, with an important and clear contribution (together with its companion manuscript that used an auditory task). My largest concerns are, perhaps, best taken as important directions for future research; though these may be worth briefly discussing in the manuscript (if authors/editors agree).

Major

The demonstration of oculomotor freezing during somatosensory temporal expectation suggest that this phenomenon may not serve the purpose of “visual stabilisation”, but may instead reflect some more global correlate of temporal expectations. This leaves the reader wondering: is there nothing “visual” about this phenomenon; or would the phenomenon be even more pronounced if there would be a clear visual benefit of stability (in some direct comparison to some equivalent visual task)? Likewise, would the link between microsaccades and perception be more profound when considering vision? If the phenomenon is completely a-modal (which would require comparing modalities directly, ideally within the same experiment) then this would make a stronger case, as it would suggest more definitely that this is not about visual stability, but instead reflects a purely global correlate of temporal expectations.

Relatedly, what might such a global correlate reflect; what might underlie it? The possibility of “alertness” is only mentioned in passing. However, this appears a likely candidate to me; also given prior studies linking such freezing to “task demands”. In future studies, it would be worthwhile manipulating alertness/arousal (e.g., through difficulty) and temporal expectations orthogonally as one way to get at this question. Another way would be to look at complementary physiological measures such as pupil size. Could the authors not also look at pupil size here? Does this also vary with expectations and with performance? Are micro-saccade rates and pupil size related?

Also, in account 1 (introduction), it is stated that freezing may be a byproduct of temporal expectations that serves no perception purpose and that, under this account, it should not be correlated with performance. I disagree. Given the correlational nature of this link, I do not see why a “byproduct” could not be “correlated” with performance, as long as the underlying cause (e.g., alertness) has an influence on both. This should probably be revised; and the notion that this may reflect a “byproduct” could possibly also be raised again in the discussion. I do not think it makes the results less interesting; it is just something that requires further experimentation.

Other

*One interesting observation regards the pro-active generation of microsaccades in anticipation of upcoming freezing (a stronger “rebound” to the cue when inhibition is expected to be needed early). This may indeed reflect a pro-active mechanism. However, it may also be due to the fact that in regular short-SOA blocks, the pace of the task is much faster? Have the authors considered looking at cue responses in irregular blocks when the preceding trial was short vs. long? If this would show the same cue-related response (larger rebound when previous was short) then this may suggest it is not a pro-active effect, but a re-active trial-history effect.

*The first paragraph of the introduction explicitly refers to theoretical scenarios that involve results from both the somatosensory (current) and the auditory (companion) manuscripts. This is a bit odd; if the comparison between somatosensory and auditory is so important; why not put them together in a single manuscript? At this stage, however, I would not want to urge the authors to merge their manuscripts. Instead, I would propose to rewrite the first paragraph to make it clear how this manuscript on somatosensory anticipation makes an important point as a stand-alone paper. (It is still

useful to cross-reference to the auditory companion article of course).

*Fig 4C appears to show the data collapsed over regular and irregular conditions. Given the large behavioural differences between these conditions, the authors may wish to consider plotting the effects for both conditions separately (just like in 4A).

*Because difficulty was continuously adjusted throughout the experiment, how can we relate microsaccades to accuracy properly? Could this related be mediated by actual stimulus intensity? Did the authors try correlating performance to actual (trial wise) stimulus intensity (and/or regress it out from their analyses)?

*Does behavioural performance also depend on microsaccade direction? Is detriment larger when microsaccades are in the direction away from the stimulated hand?

Minor

*How did participants respond (I may have overlooked this)?

*Accuracy rule set to 71%, but the data show performance fluctuations around 65%. How can this be explained?

*The discussion brings up the distinction between attention vs expectation; and states how current findings must reflect expectation. However, as the targets were always task-relevant here, I am not sure how well the distinction is justified here.

*It would be interesting to know whether there is also an effect of “surprise” on microsaccades. Could the authors look at the microsaccade rate after early targets when these were expected (regular) vs not (irregular).

*Another seemingly relevant paper for oculomotor freezing with temporal expectations that the authors may have overlooked is: Olmos-Solis, K., van Loon, A. M., Los, S. A., & Olivers, C. N. (2017). Oculomotor measures reveal the temporal dynamics of preparing for search. In *Progress in brain research* (Vol. 236, pp. 1-23).

Signed: Freek van Ede

REVIEWER #1

This is an interesting manuscript, probably deserving high-impact publication. They that tactile frequency discriminations are accompanied by a slowing of micro-saccades, especially when the stimulus is predictable. I think this is important for several reasons, not least that micro-saccade “freezing” could become a marker for perceptual expectancy. As far as I can tell, the experiments have been conducted with psychophysical rigor, and the analysis appropriate. I make a couple of comments, but in general recommend publication.

As far as I can understand, this is one manuscript of a pair, both showing cross-modal action of micro-saccade slowing, one for touch the other for audition. While I approve high-impact publication of one, both are not high-impact. Either the authors should combine both into a single paper (condensing much of the redundancies), or they publish one as the high-impact, and the other as a follow-up, confirmation with incremental added value, showing the same phenomenon occurs also in audition.

In any event, the manuscript would profit from being condensed, removing repetition and tightening the language. I think a shorter manuscript would be more impactful.

Other than that, nice work!

— Many thanks for the positive evaluation. When possible, we have tightened the language of our manuscript in the revision. We have also deleted several paragraphs [e.g., lines 450-455; lines 498-504 in the original manuscript].

REVIEWER #2 (Signed: Freek van Ede)

The authors investigated the phenomenon of “oculomotor freezing” (inhibition of micro-saccades) as a correlate of temporal expectations. Critically, in contrast to prior work on this phenomenon (most of which by the authors), such freezing was here demonstrated not in anticipation of upcoming vision, but upcoming touch (somatosensory frequency perception). This suggests that this phenomenon may not serve the purpose of visual stabilisation (as previously interpreted), but rather reflect some global correlate of a supramodal temporal expectation. In further support of this, the authors were also able to relate microsaccade rate and timing to somatosensory perception performance; with worse performance when microsaccades occurred close in time to the tactile task, and in trials in which microsaccades occurred at a higher rate.

This is a nice manuscript that is well written and structured, with an important and clear contribution (together with its companion manuscript that used an auditory task). My largest concerns are, perhaps, best taken as important directions for future research; though these may be worth briefly discussing in the manuscript (if authors/editors agree).

— Many thanks for your thoughtful comments. Addressing them has strengthened our manuscript.

Major

The demonstration of oculomotor freezing during somatosensory temporal expectation suggest that this phenomenon may not serve the purpose of “visual stabilisation”, but may instead reflect some more global correlate of temporal expectations. This leaves the reader wondering: is there nothing “visual” about this phenomenon; or would the phenomenon be even more pronounced if there would be a clear visual benefit of stability (in some direct comparison to some equivalent visual task)? Likewise, would the link between microsaccades and perception be more profound when considering vision? If the phenomenon is completely a-modal (which would require comparing

modalities directly, ideally within the same experiment) then this would make a stronger case, as it would suggest more definitely that this is not about visual stability, but instead reflects a purely global correlate of temporal expectations.

— We agree that in the future it would be very interesting to further explore the extent of the oculomotor freezing across modalities, we initially had considered such a comparison, but it is non-trivial due to the need to equate task difficulty across modalities. This comparison is beyond the scope of the present study.

Relatedly, what might such a global correlate reflect; what might underlie it? The possibility of “alertness” is only mentioned in passing. However, this appears a likely candidate to me; also given prior studies linking such freezing to “task demands”. In future studies, it would be worthwhile manipulating alertness/arousal (e.g., through difficulty) and temporal expectations orthogonally as one way to get at this question. Another way would be to look at complementary physiological measures such as pupil size. Could the authors not also look at pupil size here? Does this also vary with expectations and with performance? Are micro-saccade rates and pupil size related?

— We agree that in the future it would be interesting to manipulate alertness and to look at pupil size. Indeed, we had tried to do so for our current data, but the pupil response is very sluggish and our intertrial intervals are short. Thus, it was not possible to establish a stable baseline and separate the pupil's reactions to the different events within a block.

Also, in account 1 (introduction), it is stated that freezing may be a byproduct of temporal expectations that serves no perception purpose and that, under this account, it should not be correlated with performance. I disagree. Given the correlational nature of this link, I do not see why a “byproduct” could not be “correlated” with performance, as long as the underlying cause (e.g., alertness) has an influence on both. This should probably be revised; and the notion that this may reflect a “byproduct” could possibly also be raised again in the discussion. I do not think it makes the results less interesting; it is just something that requires further experimentation.

— We agree with this point; we deleted the sentence about the correlation with performance and revisited the point in the Discussion [lines 516-520].

Other

*One interesting observation regards the pro-active generation of microsaccades in anticipation of upcoming freezing (a stronger “rebound” to the cue when inhibition is expected to be needed early). This may indeed reflect a pro-active mechanism. However, it may also be due to the fact that in regular short-SOA blocks, the pace of the task is much faster? Have the authors considered looking at cue responses in irregular blocks when the preceding trial was short vs. long? If this would show the same cue-related response (larger rebound when previous was short) then this may suggest it is not a pro-active effect, but a re-active trial-history effect.

— This is an interesting idea. We have analyzed the data accordingly and found that the cue-target-interval in the previous trial did not affect the size of the rebound in irregular blocks [lines 578-580; Fig. S6].

*The first paragraph of the introduction explicitly refers to theoretical scenarios that involve results from both the somatosensory (current) and the auditory (companion) manuscripts. This is a bit odd; if the comparison between somatosensory and auditory is so important; why not put them together in a single manuscript? At this stage, however, I would not want to urge the authors to merge their manuscripts. Instead, I would propose to rewrite the first paragraph to make it clear how this manuscript on somatosensory anticipation makes an important point as a stand-alone paper. (It is still useful to cross-reference to the auditory companion article of course).

— We have revised the first paragraph [lines 37-42] and the rest of the manuscript accordingly and cross-referenced the auditory paper.

*Fig 4C appears to show the data collapsed over regular and irregular conditions. Given the large

behavioural differences between these conditions, the authors may wish to consider plotting the effects for both conditions separately (just like in 4A).

— We have replaced **Fig 4C** and now show the conditions separately.

*Because difficulty was continuously adjusted throughout the experiment, how can we relate microsaccades to accuracy properly? Could this related be mediated by actual stimulus intensity? Did the authors try correlating performance to actual (trial wise) stimulus intensity (and/or regress it out from their analyses)?

—We have tried to regress stimulus frequency out for all statistical models that relate accuracy to microsaccades. As expected, accuracy is predicted by stimulus frequency. Importantly, the effect of microsaccades preceding or accompanying the target persists when stimulus frequency is added as a predictor. Yet, the models do not converge properly, the fitting algorithm classifies the models as nearly unidentifiable. The models become too complex for our unbalanced data (few trials have a microsaccade before or during the target). Due to this residual uncertainty regarding the statistical models, we would very much prefer to not report these. Please note that the relation between microsaccades and accuracy becomes significant already before the stimulus is presented. For this reason, we are confident this relation is not driven by stimulus frequency/intensity. We now mention this point in the manuscript [**lines 396-399**].

*Does behavioural performance also depend on microsaccade direction? Is detriment larger when micro-saccades are in the direction away from the stimulated hand?

– We had looked at microsaccade direction and now report that it is not biased toward the stimulated hand [**lines 197-198; Fig. S1B**]. Moreover, behavioral performance does not depend on microsaccade direction. Now we include this analysis [**lines 281-285; lines 394-399; Fig. S5**].

Minor

*How did participants respond (I may have overlooked this)?

– We had reported that participants responded by pressing one of two buttons, each with a different finger of the dominant hand. We have rephrased the sentence [**lines 151-152**].

*Accuracy rule set to 71%, but the data show performance fluctuations around 65%. How can this be explained?

– We now acknowledge this discrepancy which we attribute to the small step size of the adaptive procedure in combination with added noise from the procedure we used [**lines 301-303**].

*The discussion brings up the distinction between attention vs expectation; and states how current findings must reflect expectation. However, as the targets were always task-relevant here, I am not sure how well the distinction is justified here.

– We have further clarified this distinction [**lines 452-455**]. Expectation, i.e., the probability of the target occurring at specific points in time was manipulated between the regular and irregular conditions, but attention which corresponds to target relevance was not manipulated, it was constant across conditions.

*It would be interesting to know whether there is also an effect of “surprise” on microsaccades. Could the authors look at the microsaccade rate after early targets when these were expected (regular) vs not (irregular).

– Please note that we have already looked at this comparison (see **Fig. 3C**, post-target interval]. We found no indication of surprise effects: no significant change in microsaccade rates and their dependence on foreperiod as well as predictability condition before and after the target (i.e., no significant interaction effects involving the factor interval, **Table S2**, right column).

*Another seemingly relevant paper for oculomotor freezing with temporal expectations that the authors may have overlooked is: Olmos-Solis, K., van Loon, A. M., Los, S. A., & Olivers, C. N.

(2017). Oculomotor measures reveal the temporal dynamics of preparing for search. In Progress in brain research (Vol. 236, pp. 1-23).

- We had not referred to this study because the authors did not manipulate predictability directly, as they did not include unpredictable targets. Nevertheless, we now cite this study [**line 528**].

****REVIEWERS' COMMENTS:**

Reviewer #2 (Remarks to the Author):

The authors have adequately addressed my comments (most of which were suggestions for future research anyways), and have used them to further improve the quality of their report.

Reviewer #2 (Remarks to the Author):

The authors have adequately addressed my comments (most of which were suggestions for future research anyways), and have used them to further improve the quality of their report.

- We thank the reviewer for the positive evaluation of our manuscript and the interesting suggestions for future research.